# The Penn Medicine BioBank: Towards a Genomics-Enabled Learning Healthcare System to Accelerate Precision Medicine in a Diverse Population

**DOI:** 10.3390/jpm12121974

**Published:** 2022-11-29

**Authors:** Anurag Verma, Scott M. Damrauer, Nawar Naseer, JoEllen Weaver, Colleen M. Kripke, Lindsay Guare, Giorgio Sirugo, Rachel L. Kember, Theodore G. Drivas, Scott M. Dudek, Yuki Bradford, Anastasia Lucas, Renae Judy, Shefali S. Verma, Emma Meagher, Katherine L. Nathanson, Michael Feldman, Marylyn D. Ritchie, Daniel J. Rader, For The Penn Medicine BioBank

**Affiliations:** 1Department of Medicine, Division of Translational Medicine and Human Genetics, University of Pennsylvania, Philadelphia, PA 19104, USA; 2Institute for Translational Medicine and Therapeutics, University of Pennsylvania, Philadelphia, PA 19104, USA; 3Department of Genetics, University of Pennsylvania, Philadelphia, PA 19104, USA; 4Department of Surgery, Division of Vascular Surgery and Endovascular Therapy, University of Pennsylvania, Philadelphia, PA 19104, USA; 5Department of Pathology, University of Pennsylvania, Philadelphia, PA 19104, USA; 6Department of Psychiatry, University of Pennsylvania, Philadelphia, PA 19104, USA; 7Abramson Cancer Center, Perelman School of Medicine, University of Pennsylvania, Philadelphia, PA 19104, USA

**Keywords:** genomics, electronic health records, biobank, PMBB, precision medicine, learning health system

## Abstract

The Penn Medicine BioBank (PMBB) is an electronic health record (EHR)-linked biobank at the University of Pennsylvania (Penn Medicine). A large variety of health-related information, ranging from diagnosis codes to laboratory measurements, imaging data and lifestyle information, is integrated with genomic and biomarker data in the PMBB to facilitate discoveries and translational science. To date, 174,712 participants have been enrolled into the PMBB, including approximately 30% of participants of non-European ancestry, making it one of the most diverse medical biobanks. There is a median of seven years of longitudinal data in the EHR available on participants, who also consent to permission to recontact. Herein, we describe the operations and infrastructure of the PMBB, summarize the phenotypic architecture of the enrolled participants, and use body mass index (BMI) as a proof-of-concept quantitative phenotype for PheWAS, LabWAS, and GWAS. The major representation of African-American participants in the PMBB addresses the essential need to expand the diversity in genetic and translational research. There is a critical need for a “medical biobank consortium” to facilitate replication, increase power for rare phenotypes and variants, and promote harmonized collaboration to optimize the potential for biological discovery and precision medicine.

## 1. Introduction

Precision medicine incorporates clinical, environmental, lifestyle, family, and genomic data to tailor disease management and optimize disease prevention and health maintenance. Since the completion of the human genome project, large-scale genomic information linked to individual-level phenotype data has fueled biomedical discovery and has become an integral component of precision medicine. Large amounts of clinical data are generated daily in clinical care and stored in the electronic health record (EHR). Linking the phenomic data encompassed in the EHR with biospecimens and genomic data from appropriately consented individuals at scale represents a tremendous opportunity for biomedical discovery and precision medicine. Penn Medicine, part of the University of Pennsylvania, is a large integrated academic health system with six hospitals and ten multispecialty centers that serve south-central Pennsylvania, south-central New Jersey, and northern Delaware. The Penn Medicine BioBank (PMBB) was launched with the intent of harnessing clinical data for discovery, creating a genomics-enabled learning healthcare system [1], and facilitating precision medicine for disease prevention and personalized therapy. Patients are enrolled under a single IRB-approved protocol that enables the acquisition of biospecimens, generation of genomic and multi-omic data, linkage to clinical information included in the EHR, and permission to recontact participants for future studies and/or the return of clinically relevant results. The scientific goal of the PMBB is to promote the integration of clinical and genomic data to power biomedical discovery and precision medicine. This report describes the operations and architecture of the PMBB and summarizes the information on the first ~170,000 participants recruited.

## 2. Materials and Methods

### 2.1. Planning and Development of PMBB

Recognizing the need for access to large numbers of appropriately consented and well-characterized human biospecimens to conduct translational research, in 2008, Penn Medicine established an IRB protocol and process for consenting patients and obtaining blood for genomic and biomarker research. In 2013, after a strategic planning process identified a pressing institutional mandate for an expanded biobank resource, the Penn Medicine BioBank (PMBB) was formally constituted, funded, and launched under the Institute of Translational Medicine and Therapeutics (ITMAT) in order to ensure it was integrated with critical infrastructure relevant to clinical and translational research and precision medicine.

The PMBB protocol was established as an institutional umbrella protocol under which any registered patient of Penn Medicine aged 18 or older was eligible, with no exclusions except an inability to provide informed consent. The core features of the consent include: (1) provision of a blood sample for biobanking and broad use for data generation, including genomic data and permission to bank any other residual tissues obtained in the context of clinical care; (2) permission to access data from the EHR for the purpose of research; and (3) permission to recontact participants for potential future studies or to return results.

### 2.2. Patient-Participant Recruitment

Initially, PMBB enrollment was accomplished through face-to-face encounters with clinical research coordinators (CRCs) in outpatient clinical areas, prioritizing locations where procedures that involved access to blood samples (phlebotomy labs, imaging procedures involving IV placement, cardiac catheterization labs, etc.) were being performed. After the onset of the COVID-19 pandemic, in August 2020, the PMBB transitioned to remote recruitment efforts to prioritize patient-participant and staff safety by initiating an electronic consent and enrollment process through REDCap, a secure web platform for building and managing online databases and surveys. Simultaneously, a process for consent utilizing the EHR (PennChart, Epic) was developed, which was initially done in person at the time of patient check-in, and then also expanded to include consent at the time of pre-check-in through their myPennMedicine online patient portal, available through web browsers and mobile devices. Eligible patients scheduled for an upcoming outpatient office visit at one of the UPHS clinic sites actively recruiting (Figure 1E) receive the PMBB consent form as part of their electronic pre-visit check-in process and have the option to complete the form online through this portal. The three-page consent form includes a link to the PMBB website, the PMBB email address, and the PMBB enrollment telephone number, which is staffed by CRCs on weekdays from 7:30 a.m. to 5 p.m. local time to answer questions potential participants may have as they are completing the consent procedure. A small percentage (~5%) of patients who either are not eligible to receive the online pre-check-in (e.g., certain surgical patients), or patients who skip the consent form during their online pre-check-in, are consented in person by registration desk staff when the patient physically reports for their appointment. PMBB brochures, with basic information about the PMBB, the PMBB website link, and contact information, are also available for registration desk staff to distribute to patients during the consent process. The website also contains a short video for patients explaining how PMBB participants contribute to scientific research.

### 2.3. Sample Collection

The PMBB patient participants consent to the collection of identifying information (e.g., name, date of birth, medical record number, and contact information), information from medical records (e.g., test results, medical procedures, medical diagnosis and procedure codes, lab values, images such as X-rays, and medicines), blood samples (up to four tablespoons), urine, saliva and/or respiratory specimens, and residual samples from clinical pathology. There is no limit on the length of time samples may be kept in the biobank. All blood samples are banked as whole blood, plasma, serum, buffy coat, and DNA for future studies following stringent standard operating procedures. Samples are collected in sterile vacutainer tubes barcoded with an identification number and scanned into a system for sample tracking. Using the institution’s adopted laboratory information system, LabVantage (LabVantage Solutions, Inc., Somerset, NJ, USA), biospecimens are processed and tracked with time-date stamps to document processing and freezing times throughout the laboratory workflow. Sample inventory is robustly supported with real-time, adaptive sample pull lists and images of sample pull locations, as well as simultaneous creation of distribution boxes and decrement of sample aliquots.

Prior to the COVID-19 pandemic, blood samples were collected from patients at the point of enrollment by CRCs cross-trained as certified phlebotomists. With the transition to electronic consenting, the consenting and sample collection processes have been decoupled. For sample collection, an automated weekly report is generated containing a list of consented patients for whom a blood sample is absent who have an upcoming phlebotomy appointment the following week at select Penn Medicine sites. Every Friday, three PMBB physicians place the electronic lab order for PMBB blood draws in the EHR of these patients. When they report to their phlebotomy appointment, the phlebotomist adds on the PMBB blood order to the patient’s existing orders and collects the blood sample. One 6 mL EDTA tube is collected per patient. All blood samples are transported to the centralized clinical laboratory and logged in prior to being transferred to the PMBB core laboratory, where the blood is processed and stored following standard operating procedures.

Additionally, the PMBB banks residual biospecimens and tissues (for example, blood, urine, cerebrospinal fluid, or tissue collected as part of clinical care) when available as fresh, frozen or fixed dependent upon the tissue histology, following standard procedures. Residual tissues are released by the Department of Pathology after examination if the specimen or tissue is determined to be in excess of that required for patient care, or for tissue or bodily fluid that would otherwise be waste material.

### 2.4. Genomic Data Generation

#### 2.4.1. Genotyping and Imputation

DNA samples on approximately 44,000 PMBB participants have been genotyped to date on an Illumina Global Screening Array v.2.0 (GSAv2) by the Regeneron Genomics Center (RGC). The genotyping array chip has a backbone of 654,027 genetic markers as well as additional ancestry informative markers. In addition, approximately 8595 of the 44,000 PMBB participants were also genotyped in the Center for Applied Genomics (CAG) at the Children’s Hospital of Philadelphia on the GSAv1 and GSAv2 genotyping array. After performing sample-level quality control (QC), genotype imputation was performed using Eagle v2.4.1 [2] and Minimac4 version 1.0.0 [3] software on the TOPMed Imputation Server [3]. Imputation was performed for all autosomes, with TOPMed version R2 on a GRCh38 reference panel. Cosmopolitan post-imputation QC included imputation score filtering (R2 > 0.7), removal of palindromic variants, biallelic variant check, sex check, genotype call rate (>99%) and sample call rate (>99%) filtering, minor allele frequency filtering (MAF > 1%), and a Hardy–Weinberg equilibrium test (*p*-value > 1 × 10^−8^). We generated PCAs to adjust for population structure and to identify genetically informed ancestry (GIA) from EIGENSOFT version 7.2.0 [4].

#### 2.4.2. Whole Exome Sequencing

Whole exome sequencing (WES) has been performed on approximately 44,000 participants to date by the RGC. DNA samples were processed with the custom IDT xGen v1 exome capture platform and sequenced on the Illumina NovaSeq 6000 (Albany, NY, USA) system on S4 flow cells. Sequence alignment, variant identification, and genotype assignment were performed using a WeCall variant caller. Sample level QC steps were then applied and sample sex errors, high rates of heterozygosity/contamination (D-stat > 0.4), low sequence coverage (less than 85% of targeted bases achieving 20X coverage), or genetically identified sample duplicates, were excluded. Additional filters were applied to pVCFs. Any SNV genotype with a read depth of less than seven reads (DP < 7) was changed to a no-call. After the application of the DP genotype filter, only the SNV variant sites that met at least one of the following two criteria were retained: (1) at least one heterozygous variant genotype with an allele balance ratio greater than or equal to 15% (AB ≥ 0.15); (2) at least one homozygous variant genotype. The same filtering was applied to INDEL variants but with an INDEL depth filter of DP < 10 and an INDEL allele balance cutoff of AB >= 0.20. Multi-allelic variant sites in the PVCF file were normalized by left-alignment and represented as bi-allelic. The variant frequency data for exome sequences and imputed data are available here: https://pmbb.med.upenn.edu/allele-frequency (accessed on 18 November 2022). The PMBB has a return of actionable results program for genomic findings that have a potential impact on participant health; this program is beyond the scope of this manuscript and will be described in a separate manuscript.

### 2.5. Clinical Data and Clinical Informatics Core

Clinical data are obtained through multiple sources, including a questionnaire completed at the time of recruitment and the Penn Medicine Clinical Data Warehouse and PennG&P (Penn Genotype & Phenotype) platform. PennG&P contains over 5.6 million patient records and other discrete clinical information amalgamated from 12 different source systems throughout the enterprise. PennG&P is based on a standard research data model called the Observational Medical Outcomes Partnership (OMOP), Common Data Model (CDM) [5], which is used worldwide by the Observational Health Data Sciences and Informatics (OHDSI) research consortium. It uses standardized language from national coding systems, such as SNOMED, LOINC [6], and RxNorm [7], for consistent terms and the labeling of information. Additionally, the PMBB maps International Classification of Diseases (ICD)-9 and ICD-10 codes to 1866 discrete disease traits using Phecode groupings [8].

### 2.6. Access to Data and Biospecimens

In keeping with the expansive mission of the PMBB, data and biospecimens are available to investigators throughout the Perelman School of Medicine and Penn Medicine for a broad range of research. External collaborations, including those with other academic institutions as well as biopharma, are encouraged and proceed through scientific collaboration with identified local Penn investigators. All research studies must have study specific IRB approval because the umbrella PMBB IRB protocol covers only sample and data acquisition, processing, storage, and dissemination. Researchers request access to data and biospecimens using a simple REDCap project proposal intake form which is then reviewed by the PMBB Steering Committee. Proposals are evaluated for scientific merit and priority, as well as the efficient use of data and biospecimens, as well as the ability to impact the care provided by Penn Medicine clinicians. The PMBB Steering Committee provides feedback to the investigator with either approval to move forward or with concerns to be addressed. This REDCap project also includes a Data Access Agreement form that must be signed by investigators prior to gaining access to any PMBB data. Per the terms of this agreement, the sharing of PMBB data with additional collaborators, whether they are internal or external to Penn, must be handled by the PMBB and not the investigators themselves, to maintain the confidentiality and integrity of any protected health information (PHI) included within PMBB datasets.

Standardized EHR clinical data are deidentified and provided to approved investigators in a secure computing environment. For assistance with the creation of more complicated phenotypes, researchers have access to the Clinical Informatics Core (CIC), a shared resource that is managed by the Institute for Biomedical Informatics in collaboration with the PMBB. The CIC is staffed by clinical data scientists with expertise in data extraction, natural language processing (NLP), and data analysis.

## 3. Results

### 3.1. Enrollment

During the initial phase of recruitment from 2008 to 2013, 13,366 Penn Medicine patients were enrolled (Figure 1A,B). In 2013, recruitment was expanded, resulting in a steady increase in enrollment to ~71,000 participants by the end of 2019 (Figure 1A,B). In 2020, the transition to electronic consenting triggered a rapid expansion in PMBB enrollment, with the total number of participants more than doubling between 2020 and 2022. There were 174,712 total participants enrolled in the PMBB as of September 2022 (Table 1, Figure 1A,B). Presently, nearly 3500 new participants are being enrolled weekly across two Penn Medicine hospitals that are actively recruiting through all their ambulatory sites, and recruitment at the other four Penn Medicine hospitals is targeted to begin in 2023. The PMBB currently has obtained and processed blood biospecimens from approximately 50% of enrolled participants; the recent shift to an electronic consent process has resulted in enrollment outpacing sample collection (Figure 1B). Active processes are underway to obtain biospecimens on the remainder of enrolled individuals. The goal is to enroll > 1 million Penn Medicine patient participants, with >90% providing a blood sample for DNA and biomarker studies.

The PMBB participant population currently represents approximately 2.5% of active Penn Medicine patients. Similar to the general Penn Medicine patient population, a slightly higher percentage of PMBB participants are female (55.9%) as compared to male (44.1%) (Table 1, Figure 1C). The age distribution of PMBB participants also tracks with that of Penn Medicine patients, with participants ranging in age from 18 years to >100 years (Table 1 and Figure 1C). The age distributions differ slightly between sex, with males trending towards an older age (Figure 1C). The PMBB cohort is diverse: 17% of enrolled PMBB participants (and 25% of genotyped/sequenced participants) are identified as African American, 4% as Asian, and 3.3% as Hispanic (Table 1; Figure 1D). With nearly 30,000 African-American patient participants currently enrolled, the PMBB has, to our knowledge, the largest number of African-American participants of any single-institutional medical biobank in the US. As shown in Figure 1E, most biobank participants reside within the Philadelphia metropolitan area including New Jersey and Delaware (53.6%); there is also a total of 3.3% of participants from other states across the US.

### 3.2. Clinical Data Availability

There are over 66.7 million data points collected in the form of encounters, diagnosis codes, procedure codes, and medication orders (Table 2). Across the PMBB cohort, over 46.7 million encounters, 10 million diagnosis codes, 3.6 million procedure codes, and 6.3 million medication orders have been recorded, averaging 268 encounters, 57 diagnosis codes, 21 procedure codes, and 36 medication orders per individual (Table 2). The most common diagnoses codes include hypertension, hypercholesterolemia, obesity, and insomnia (Figure 2).

### 3.3. Phenome-Wide Association Study (PheWAS) of BMI

To evaluate the validity of the PMBB clinical data, a phenome-wide association study (PheWAS) was conducted using mean body mass index (BMI) as the exposure and 1856 disease traits derived from grouping encounter diagnoses using phecodes as the outcome. All the models were adjusted with age, sex, and self-reported race in the EHR. A total of 662 associations of BMI with at least one disease trait across the 18 disease categories passed Bonferroni correction for multiple hypothesis testing (*p* < 2.6 × 10^−5^). The strongest associations with BMI were with type 2 diabetes, hyperlipidemia, overweight, obesity, sleep apnea, and osteoarthritis (all *p* < 1 × 10^−331^, Figure 3, Appendix A). The associations with mean BMI show evidence of its effect on all organ systems, covering associations with the spectrum of disease categories (Figure 3). Additional associations include hypertension, heart failure, endometrial hyperplasia, bone fracture, depression, pregnancy complications, and respiratory failure (Figure 3).

### 3.4. Laboratory-Wide Association Study (LabWAS) of BMI

We extracted 24 clinical laboratory measurements from the EHR for all the PMBB participants. These lab measurements were selected based on a common lab test in the health system and contained at least 1000 individuals within each lab which was measured. We computed median values for each individual within each lab and, as a proof-of-concept, evaluated their association with BMI. Linear regression was performed to test for association and all the models were adjusted with age, sex and self-reported race. We replicated many known associations between BMI and lab values (Figure 4, Appendix A). For example, blood glucose measures were significantly associated with increased BMI. Triglyceride levels were significantly positively associated and high-density lipoprotein cholesterol (HDL-C) levels were significantly inversely associated with BMI, as expected. Markers of inflammation were also significantly associated with BMI. In this proof-of-concept analysis of the lab measurements, the associations with BMI support the association with the disease outcomes reported in the PheWAS.

### 3.5. Genome-Wide Association (GWAS) with BMI

As a proof-of-concept, we conducted a GWAS for median BMI within five genetically inferred ancestry groups. These included 30,360 EUR, 11,300 AFR, 711 AMR, 680 EAS, and 573 SAS individuals in the PMBB (Table 1). The analysis tested the association of ~7.6 million SNPs with MAF > 1%, imputation R2 > 0.3, using a linear mixed model implemented in REGNIE. All the models were adjusted with age, sex, and the first six ancestry specific principal components to account for population stratification. We then conducted cross-ancestry meta-analysis by integrating GWAS summary statistics from each ancestry group using PLINK. Our meta-analysis identified 201 genome-wide significant SNP associations with BMI (*p* < 5 × 10^−08^, Figure 5), replicating several previously reported associations in published GWAS of BMI. The strongest association in our PMBB analysis was with *FTO* variant rs55872725 (*p* = 4.7 × 10^−28^, beta = 0.271), which has been previously reported. Other known associations included rs7559547 (*p* = 3.92 × 10^−14^, beta = 0.41, *TMEM18*) and rs539515 *(p =* 9.02 × 10^−11^, beta = 0.36, *SEC16B*), among others. Summary statistics of results with *p* < 1 × 10^−04^ are available in Appendix A.

## 4. Discussion

The Penn Medicine BioBank was created to harness clinical data and biospecimens for biomedical research within Penn Medicine, a large academic healthcare system. Within a decade, it has emerged as a critical resource for translational medicine that has fueled discovery science and facilitated precision medicine, empowering a genomics-enabled learning healthcare system. As of September 2022, the PMBB had enrolled over 174,000 participants, obtained biospecimens on >70,000 participants, and generated genomic data on ~44,000 of its participants. The PMBB intends to enroll >1 million participants, obtain biospecimens on >90% of participants, and generate genomic data on all participants for whom biospecimens have been obtained.

The 2015 Institute of Medicine (now National Academy of Medicine) report on Translating Genomic-Based Research for Health [1] advocated for the development of ‘genomics-enabled learning healthcare systems’ based on the systematic summarized collection and use of genomic data, integrated with phenotypic data, to make discoveries and enhance healthcare in large healthcare systems. More recently, in its strategic vision for genomics research and application of genomics to clinical care, the National Human Genome Research Institute (NHGRI) emphasized the design and implementation of genomics-enabled learning healthcare systems to include infrastructure, resources, and technology development for genomics; the inclusion of underrepresented and minority groups to make genomic research more equitable; the development of multi-omics studies to get a comprehensive view of disease biology and the progression of diseases; and building tools to implement the knowledge back into the EHR to improve healthcare [9]. Large disease-agnostic and diverse medical biobanks at academic medical centers, such as the PMBB, are a critical component of fulfilling this vision.

Despite recapitulating health and disease traits from structured diagnosis codes, and the successful integration of this with genomic data [10,11,12,13,14,15,16], it must be acknowledged that diagnosis codes are crude approximations of underlying biological traits. As such, the future of EHR-empowered genomics research lies in ‘advanced phenotyping’ beyond diagnosis codes. These approaches include laboratory data, medication data, and other forms of structured data, all of which are relatively straightforward to access and use for research. Laboratory data, procedure codes, family history, and billing codes are all being mapped to concepts from various vocabularies (MONDO [17], SNOMED) to develop phenotypic algorithms that characterize the outcome of interest. Even more exciting and informative is the extraction of meaningful quantitative information from unstructured data (e.g., clinical notes, procedure reports, imaging reports, and pathology reports) using natural language processing (NLP) methods. To this end, the PMBB has deployed NLP software (Linguamatics, Cambridge, UK) to extract phenotypes from clinical notes and other unstructured data in PMBB participants. Furthermore, there is an immense amount of clinical imaging performed in medical centers and these images are another potential source of phenotypic data, sometimes referred to as ‘imaging-derived phenotypes’ (IDPs). In several ongoing PMBB efforts, deep learning and machine learning techniques are being used to translate imaging data, such as CT, MRI, and fMRI, into quantitative IDPs to fuel new discovery.

Another approach to collecting additional phenotype and exposure data that are absent in the EHR is using participant questionnaires. During the COVID-19 pandemic, an initial COVID-19 survey [18] was deployed to PMBB participants to collect information on symptoms, co-morbidities, and outcomes related to COVID-19. As the pandemic has progressed, we now administer an active longitudinal survey to follow participants for up to 18 months from their first COVID-19 diagnosis, yielding insights into long COVID. Combining survey results with biospecimen and EHR-derived phenotypes can shed light on factors that predict the onset of disease, refine preventative care, and optimize the clinical trial design. Current efforts are focused on extending active data acquisition through integrating mobile devices for both real-time data collection from survey questions and biometric activity data.

A major focus of biobank research is the use of genomics to understand the genetic architecture of health and disease and its implications for clinical care by linking phenomic efforts with genomic data obtained from biospecimens. Leveraging these approaches, the PMBB has developed a robust and diverse genomics research enterprise. Studies using PMBB data have highlighted the utility of the ‘genome-first’ approach’s utility in studying rare variants at scale and identifying new associations between genes and disease [10,11], as well as refined the range of the phenotypic presentation of individuals carrying rare impactful variants in known disease genes [12,19].

To support equitable genomic research, a commitment to participant diversity has been a hallmark feature of the PMBB since its inception. Seventeen percent of PMBB participants (and 24% of those for whom biospecimens are currently available) are African Americans or immigrants of African ancestry. This diversity has led to novel genomic and biological insights that directly impact the health of underrepresented groups. For example, recent work in the PMBB highlighted that hereditary amyloid transthyretin cardiomyopathy was a common yet markedly underdiagnosed cause of heart failure among African-American individuals [20], with many cases of the disease remaining undiagnosed even at a tertiary medical center such as Penn Medicine. Given the availability of targeted therapy, this finding advocates for the aggressive utilization of genomic and precision medicine to diagnose transthyretin cardiomyopathy in this population. This ‘genome-first’ approach is revealing an under-diagnosis of other genetic conditions. A ‘return of actionable results’ program is underway, and the implications for the greater utilization of genetic testing in clinical practice are clear.

The integration of genomic data into clinical practice is essential for the next generation of healthcare. Penn Medicine is at the forefront of developing techniques to provide high-quality patient care based on real-world evidence and genomic discoveries [21,22]. An analysis of pharmacogenetic (PGx) variants in the PMBB concluded that anticipatory genotyping can efficiently lead to the effective communication of PGx results to patients [23]. Polygenic risk scores (PRS) have been posited as a novel approach to leverage common-variant genetics for clinical care to predict the genetic risk for complex diseases, although the clinical utility of this approach has yet to be fully determined [24,25]. Researchers utilizing PMBB data reported that PRS for psychiatric disorders [26] and substance use disorders [27] has shown cross-trait associations beyond traditional diagnostic boundaries, suggesting broad effects of genetic liability for these disorders. Furthermore, PRS in PMBB participants significantly increased the ability to identify the cancer status of European individuals but not African Americans, underscoring the need for large-scale genomic studies on non-white populations [13].

A critical feature of the PMBB is the availability of stored plasma and serum for biomarker analyses and integration with clinical and genomic data. Although the effort and expense of generating and storing plasma/serum aliquots are substantial, the benefit of this approach is rapidly becoming apparent. Multiple investigators have utilized the genomic data to identify PMBB participants with genotypes of interest to then use stored samples in cases and controls to measure biomarkers of interest. During the COVID-19 pandemic, access to stored samples from PMBB participants enrolled pre-pandemic who subsequently developed COVID-19 permitted investigators to address a number of important research questions [28,29,30,31]. Now, with over seven years of median follow-up data since the recruitment of PMBB patient participants, the stored samples are increasingly precious for their use in assaying biomarkers that may be predictive of incident disease. As methods for large-scale proteomics and metabolomics improve and the costs come down, the opportunity to generate plasma-based large-scale omics and integrate with genomics and clinical data is increasingly feasible and promises to further enhance biological discovery and precision medicine.

All PMBB participants consent to be recontacted, a critical feature of the protocol that is useful for several purposes. Patient-reported surveys represent an important addition to the EHR data for certain phenotypes as well as lifestyle and exposure data. Permission to recontact facilitates ‘recall-by-genotype’ deep phenotyping studies, which represents a tremendous opportunity for ‘genome-first’ discovery. Several investigators are actively performing studies in which the genomic data are used to identify individuals that carry rare variants in genes of interest or have a high polygenic risk for certain conditions, and participants are contacted to consider participation in hypothesis-driven deep phenotyping studies. Deep phenotyping can include targeted imaging, immunological profiling, provocative testing (e.g., oral glucose or fat tolerance test), creation of induced pluripotent stem cells (iPSCs), or any number of other clinical phenotyping approaches driven by the specific scientific question. Finally, the era of precision medicine will certainly include many clinical trials that are targeted to individuals of a specific inherited genotype; large medical biobanks with pre-existing genomic data, such as the PMBB, offer a fertile opportunity for the recruitment of individuals for such genotype-directed clinical trials.

## 5. Conclusions

The PMBB is a disease-agnostic institutional biobank under a single umbrella protocol based at a large academic health system with the purpose of promoting a genomics-enabled learning healthcare system to fuel scientific discovery, translational science, and precision medicine. A comprehensive biobank of DNA, plasma, and serum on all participants with selected other specimens and tissues on a subset of participants is linked to rich EHR clinical data, imaging, and survey data. The clinical database is standardized to OMOP and contains demographic, diagnoses (e.g., ICD-9/ICD-10 codes), procedures (e.g., Current Procedure Terminology—CPT codes), laboratory data, medication data, encounter types, socioeconomic factors, and survey data. The initiation of e-consenting has led to a substantial increase in the rate of enrollment. As of September 2022, genome-wide genomic data have been generated on ~44,000 participants and plasma multi-omics data on several thousand participants. The substantial representation of African-American patient participants in the PMBB addresses the urgent need to increase diversity in human genetic studies. Researchers with approved IRB protocols can request access to biobank samples and data through a data access portal. Publications supported by PMBB data and specimens can be found here: https://pmbb.med.upenn.edu/pmbb/publications.html (accessed on 18 November 2022). The PMBB is one of several large medical biobanks at academic medical centers in the US and is strongly supportive of the creation of a ‘medical biobank consortium’ to facilitate replication, increase power for rare phenotypes and variants, and promote harmonized collaboration around genotype-directed deep phenotyping and recruitment into clinical trials.

## Figures and Tables

**Figure 1 jpm-12-01974-f001:**
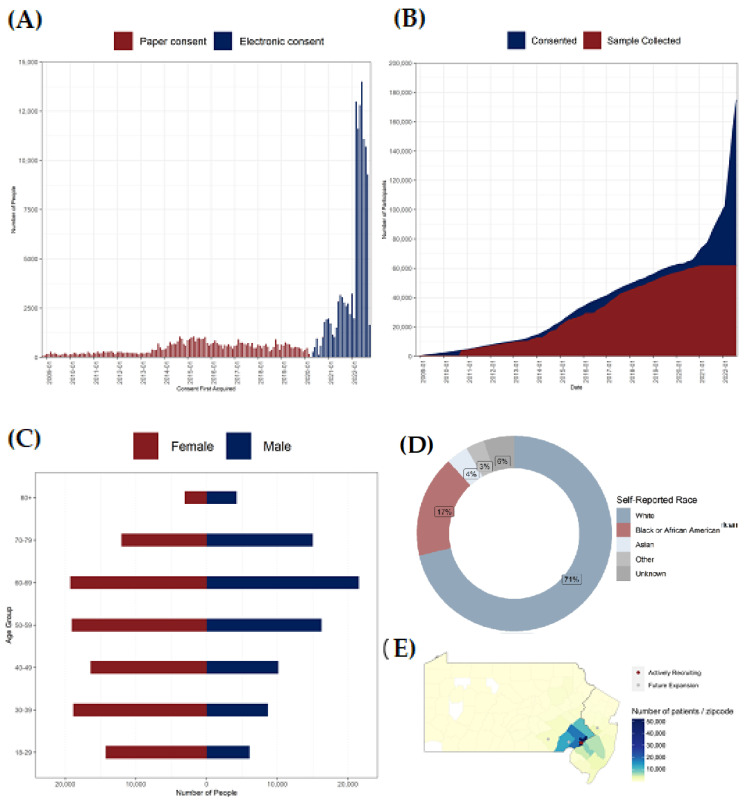
Recruitment and Demographics. (**A**) Distribution of enrollment through paper and electronic consent. (**B**) Cumulative numbers of participants consented and biospecimen sample collection. (**C**) Distribution of participants by age group and self-reported sex. (**D**) Distribution of participants by self-reported race. (**E**) Density of recruitment around the six clinical sites of UPHS in Pennsylvania and New Jersey: Hospital of the University of Pennsylvania, Penn Presbyterian Medical Center, Pennsylvania Hospital, Chester County Hospital, Lancaster General Health, Princeton Health.

**Figure 2 jpm-12-01974-f002:**
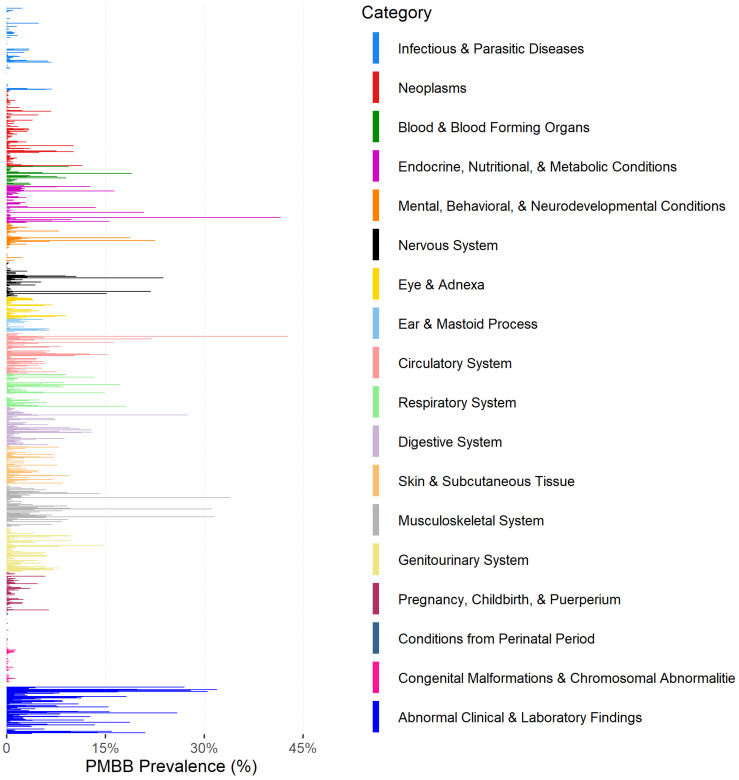
Prevalence of diagnoses code among PMBB participants grouped by broader disease domain.

**Figure 3 jpm-12-01974-f003:**
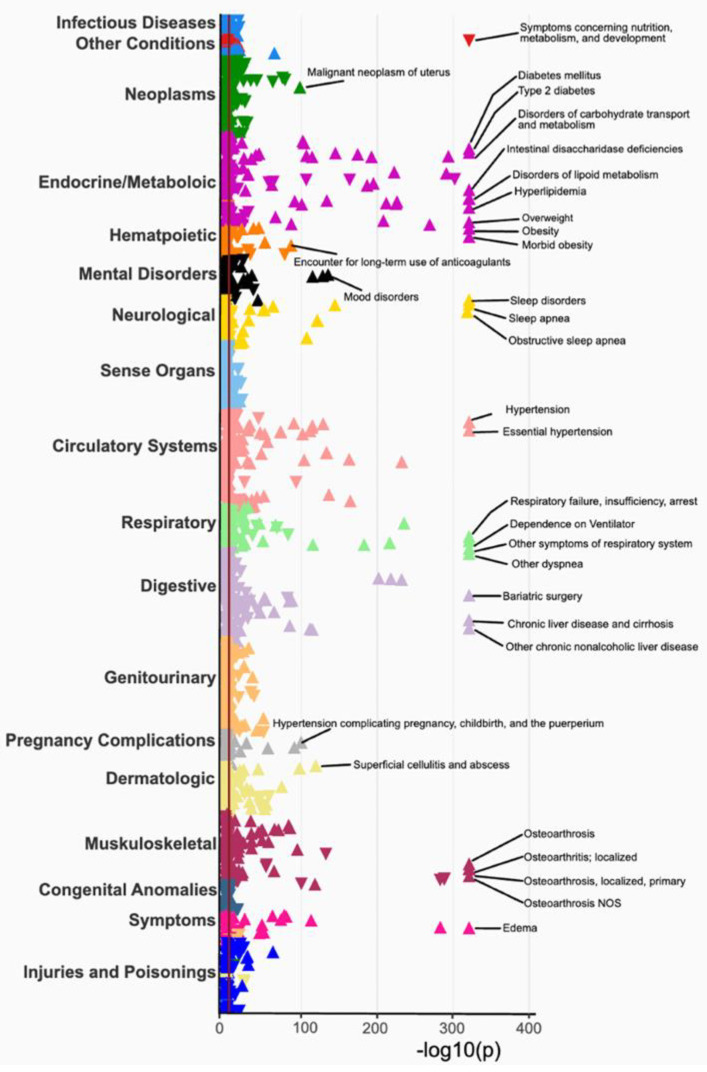
A phenome-wide association between mean body mass index and 1856 EHR-derived phenotypes.

**Figure 4 jpm-12-01974-f004:**
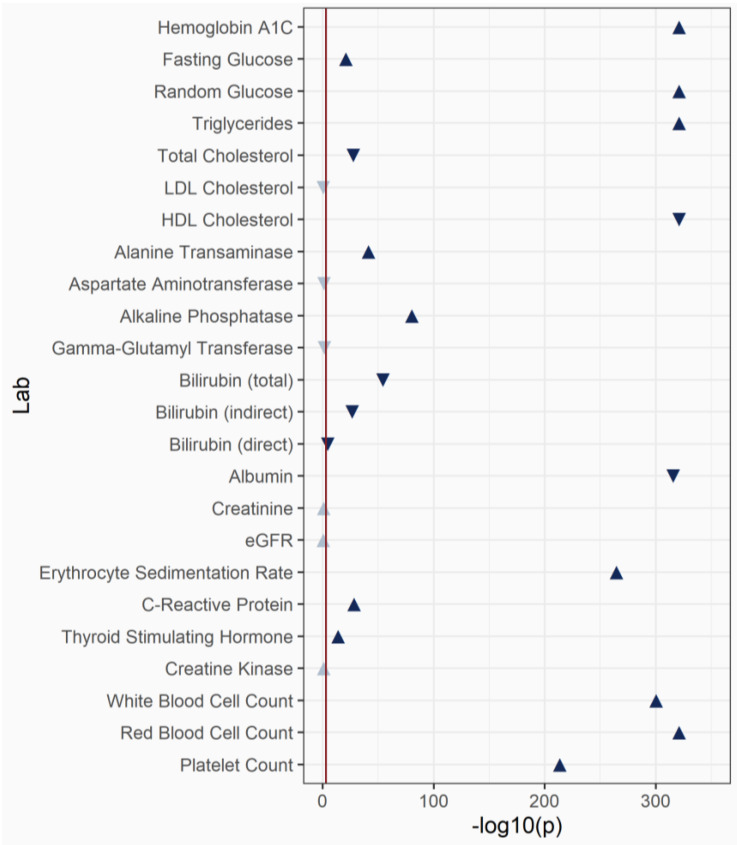
A laboratory-wide association between mean body mass index and 24 laboratory measurements derived from the electronic health records.

**Figure 5 jpm-12-01974-f005:**
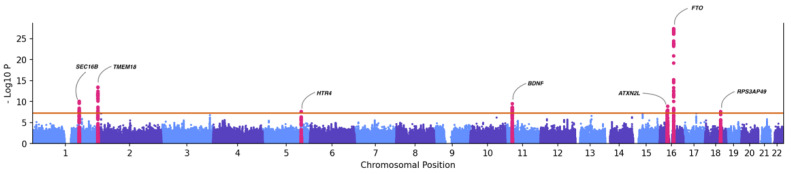
Manhattan plot showing association between common genetic variants (MAF > 1%) and BMI.

**Table 1 jpm-12-01974-t001:** Comparison of PMBB Participant Characteristics with UPHS Patient Characteristics.

	PMBB Participants n (%)	Genotyped Participants n (%)	UPHS Patients n (%) ^†^
**Total**	174,712	43,884	3,688,610
**Gender**			
Female	97,674 (55.9%)	21,965 (50.1%)	2,042,868 (55.4%)
Male	77,055 (44.1%)	21,918 (49.9%)	1,604,210 (43.5%)
Other	17 (<1%)	1 (<1%)	107 (<1%)
**Age Range (Years)**	18–103	18–103	0–121
**Age Groups**			
18–29	17,815 (10.2%)	4302 (9.8%)	392,532 (10.5%)
30–39	27,355 (15.7%)	5406 (12.3%)	518,679 (14.1%)
40–49	25,819 (14.8%)	5688 (13.0%)	430,489 (11.7%)
50–59	33,827 (19.4%)	9519 (21.7%)	458,952 (12.4%)
60–69	40,268 (23.0%)	10,839 (24.7%)	507,397 (13.8%)
70–79	28,582 (16.4%)	5941 (13.5%)	400,016 (10.8%)
80+	8811 (5.0%)	2189 (5.0%)	333,781 (9.0%)
**Self-reported Race**			
African American	29,372 (16.8%)	10,815 (24.6%)	672,461 (18.2%)
White	124,406 (71.2%)	29,329 (66.8%)	2,029,684 (55.0%)
Asian	7156 (4.1%)	979 (2.2%)	152,615 (4.1%)
Other	9386 (5.4%)	1372 (3.1%)	370,313 (10.0%)
Unknown	7499 (4.3%)	1761 (4.0%)	463,537 (12.6%)
**Self-reported Ethnicity**			
Hispanic	5715 (3.3%)	1112 (2.5%)	174,179 (4.7%)
Non-Hispanic	165,713 (94.8%)	42,425 (96.7%)	3,290,018 (89.2%)
Unknown	3284 (1.9%)	347 (0.8%)	183,723 (5.0%)
**Genetically-Inferred** **Ancestry**			
African	N/A	11,300 (25.7%)	N/A
European	N/A	30,360 (69.2%)	N/A
East Asian	N/A	680 (1.5%)	N/A
South Asian	N/A	573 (1.3%)	N/A
Admixed American	N/A	711 (1.6%)	N/A
Other	N/A	301 (0.7%)	N/A
**Median period of EHR** **follow-up since enrollment**	7 years	5.7 years	N/A

**^†^** Demographics based on EHR data on UPHS patients who have been seen at least once from 2008 to present. Data from following UPHS sites were included: Hospital of the University of Pennsylvania, Penn Presbyterian Medical Center, Pennsylvania Hospital, Chester County Hospital, Princeton Health.

**Table 2 jpm-12-01974-t002:** Clinical Data Availability for PMBB Participants.

Event Type	Total Number of Events	Mean Number of Events (SD) *
Encounter	46,738,773	268 (321)
Diagnosis Code (number of condition-related visits)	10,023,922	57.4 (58.7)
Procedure Code	3,621,056	20.7 (26.7)
Medication Order	27,914,486	159.8 (271.8)

* Average number of events each patient has for each event type.

## Data Availability

All the data used to generate the figures were made available in Appendix A.

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
