# Peer review of "The Penn Medicine BioBank: Towards a Genomics-Enabled Learning Healthcare System to Accelerate Precision Medicine in a Diverse Population"

_jpm, 2022, doi:10.3390/jpm12121974_

Round 1
Reviewer 1 Report
This is an interesting cohort profile description of the Penn Medicine Biobank, an EHR linked biobank with currently >170,000 participants recruited. I have a few major and some minor comments.
Major comments
1. Genomics is a key component of the biobank and features prominently in the title and discussion of the paper. However, the genomics data is not described at all. Only in the discussion it becomes clear that ~45,000 participants have genomics data, but it is unclear what this genomics data consists of. This is a glaring omission that needs to be fixed. Please describe in some detail what genomics data has been (and will be) collected on participants of PMBB. Is this GWAS data, whole exome sequence or whole genome sequence data? If this is eg GWAS data, details on the array used and quality control and imputation of the data need to be provided.
2. What is the procedure for external scientists to gain access to the data and/or biospecimens? Is this only possible through collaboration with local scientists? Does data access apply to all data or are there special arrangements (ie more limited access) for the genomics data? Have all genomics data been generated by Regeneron?
3. What is PBMM’s policy to return (genomics) results to participants? How do you deal with incidental findings?
4. Line 107: Is DNA extracted for all participants as part of the blood sample collection protocol?
5. Which covariates were used in the PheWAS analysis of BMI? P-values of the strongest associations all seem to be equal, which is probably a limitation (of precision) of the statistical software used. The authors may want to calculate precise p-values, which will yield more variation. It would be good if associations with heart failure, endometrial hyperplasia, bone fracture and depression mentioned in the text (line 222) are also shown in Fig. 3.
6. Suppl Table 1 was not available for me to review, the link produced an error.
7. Fig.1E mentions 6 clinical sites, but footnote of Table 1 lists only 5? Legend of Fig 1E mentions “Future expansion”, but this is not described in the main text? Furthermore, the picture is very small and details are difficult to see.
Minor comments
8. There are a number of abbreviations that need to be defined at first mention: EHR (line 19), RedCap (line 82), EMR (line 122), CSF (line 129), CPT (line 354), conversely EHR should replace the full name in lines 83 and 136.
9. Some typos etc.: line 22: participants have been; line 25: On an average; line 53: the scientific goals; line 111: closing bracket missing; line 112: stamps to document; Table 1: 70-70 = 70-79; Fig 3 legend: 1800 = 1856; line 242: genomics-enabled; line 275/6: optimize the clinical trial design; line 276: data acquisition to; line 292: hereditary; line 407: Institute of Medicine.
Author Response
Response to Reviewer 1 Comments
We are grateful to the Reviewers for their thoughtful comments. Reviewer 1 raised a few important questions that we have now addressed. Below, we provide a point-by-point response to the Reviewers’ comments.
Point 1: This is an interesting cohort profile description of the Penn Medicine Biobank, an EHR-linked biobank with currently >170,000 participants recruited. I have a few major and some minor comments.
Major comments
Genomics is a key component of the biobank and features prominently in the title and discussion of the paper. However, the genomics data is not described at all. Only in the discussion, it becomes clear that ~45,000 participants have genomics data, but it is unclear what this genomics data consists of. This is a glaring omission that needs to be fixed. Please describe in some detail what genomics data has been (and will be) collected on participants of PMBB. Is this GWAS data, whole exome sequence or whole genome sequence data? If this is eg GWAS data, details on the array used and quality control and imputation of the data need to be provided.
Response 1: Thank you for the comment. The genomic data includes imputed GWAS data and whole exome sequencing on ~44K PMBB participants. We wanted to primarily focus on the design aspect of the biobank and hence didn’t include a detailed description of the genomic data generated on the PMBB participants. Considering the reviewer's comment, we have included a new section in the Methods called “Genomic Data” (line: 144) where we have described the details requested. Additionally, we have included a summary of a genome-wide association study of BMI as a proof-of-concept study of genomic data. (Line 317)
Point 2: What is the procedure for external scientists to gain access to the data and/or biospecimens? Is this only possible through collaboration with local scientists? Does data access apply to all data or are there special arrangements (ie more limited access) for the genomics data? Have all genomics data been generated by Regeneron?
Response 2: External scientists may gain access to PMBB data and/or biospecimens through IRB-approved scientific collaborations with Penn investigators. Penn investigators requesting access to PMBB data sign the PMBB Data Access Agreement, which was created in consultation and in accordance with the University of Pennsylvania Institutional Review Board. This agreement dictates that investigators may not share any PMBB data with any collaborators, internal or external and that all data access requests must go through PMBB, in order to maintain the confidentiality of any protected health information contained within PMBB datasets. We have clarified this in lines 198-216.
Yes, the genomic data currently available has been generated by Regeneron Genetic Center, which has committed to the continued generation of genomic data. We have also independently generated array genotype data on 8,595 PMBB participants to date.
Point 3: What is PBMM’s policy to return (genomics) results to participants? How do you deal with incidental findings?
Response 3: The PMBB has a return of actionable results program for genomic results that have a direct potential impact on participant health and actionable clinical management. The details of this program are beyond the scope of this manuscript and will be described in a separate manuscript. We have added a comment about this program in the new “Genomic data” section in the Methods.
Point 4: Line 107: Is DNA extracted for all participants as part of the blood sample collection protocol?
Response 4: Yes, DNA is extracted for all participants as part of the blood sample collection protocol. We have clarified this in 122.
Point 5: Which covariates were used in the PheWAS analysis of BMI? P-values of the strongest associations all seem to be equal, which is probably a limitation (of precision) of the statistical software used. The authors may want to calculate precise p-values, which will yield more variation. It would be good if associations with heart failure, endometrial hyperplasia, bone fracture and depression mentioned in the text (line 222) are also shown in Fig. 3.
Response 5: We adjusted the regression models with age at enrollment, sex, and self-reported race in the EHR. Yes, we analyzed the data in R, and currently, base R package cannot report the values precisely below that threshold of 1e-321.
Thank you for the suggestion. We have now labeled all four conditions in figure 3.
Point 6: Suppl Table 1 was not available for me to review, the link produced an error.
Response 6: We apologize for the inconvenience. We uploaded the supplementary table to the submission portal. We have provided the file again with the revision so you should be able to review it.
Point 7: Fig.1E mentions 6 clinical sites, but footnote of Table 1 lists only 5? Legend of Fig 1E mentions “Future expansion”, but this is not described in the main text? Furthermore, the picture is very small and details are difficult to see.
Response 7: To clarify, we have updated the legend of Fig. 1E to list the names of the six clinical sites: Hospital of the University of Pennsylvania, Penn Presbyterian Medical Center, Pennsylvania Hospital, Chester County Hospital, Lancaster General Health, and Princeton Health. Table 1 does not include data from Lancaster General Health (LGH). We have also enlarged Fig. 1E to make it easier to visualize, and have added a sentence in the Results referring to “Future expansion” (line: 262).
Point 8: There are a number of abbreviations that need to be defined at first mention: EHR (line 19), RedCap (line 82), EMR (line 122), CSF (line 129), CPT (line 354), conversely EHR should replace the full name in lines 83 and 136.
Response 8: We thank the Reviewer for catching these errors, and have corrected them according to the notes below:
- EHR (line 22): expanded to “electronic health record”
- RedCap (line 88): REDCap is not an abbreviation, but the name of an online secure platform for creating surveys and maintaining databases, and we have clarified this in the manuscript.
- EMR (line 131): changed to “EHR”
- CSF (line 138): expanded to “cerebrospinal fluid”
- CPT (line 464): expanded to “Current Procedural Terminology”
- conversely EHR should replace the full name in lines 83 and 136: Replaced as suggested
Point 9: Some typos etc.: line 22: participants havebeen; line 25: On an average; line 53: the scientific goals; line 111: closing bracket missing; line 112: stamps to document; Table 1: 70-70 = 70-79; Fig 3 legend: 1800 = 1856; line 242: genomics-enabled; line 275/6: optimize the clinical trial design; line 276: data acquisition to; line 292: hereditary; line 407: Institute of Medicine.
Response 9: We thank the Reviewer for catching these errors and have corrected them.
Reviewer 2 Report
Dear Editor,
The work presented by Anurag Verma et al is a much-needed step in the field and era of personalized medicine. Most of the GWAS data collected in the past decades were from European ancestors. These data are currently used for pharmacogenomics and disease-biomarker association studies, generating a bias toward underrepresented populations. Instead, the Penn Medicine BioBank here described consists of approx 30% of participants of non-European ancestry out of 174,712 participants. Although other projects are ongoing to increase diversity in human genetic studies (like All of Us, Uk Biobank, Genome Asia, H3Africa, and TOPMed), there is still so much to be done in order to compensate for the absence of diversity. Anurag Verma et al work has a high representation of African-American patient participants in and addresses the urgent need to improve the quality of health care for all.
My advice is to publish the paper in its current form
Best regards
Author Response
Response to Reviewer 2
Point 1: The work presented by Anurag Verma et al is a much-needed step in the field and era of personalized medicine. Most of the GWAS data collected in the past decades were from European ancestors. These data are currently used for pharmacogenomics and disease-biomarker association studies, generating a bias toward underrepresented populations. Instead, the Penn Medicine BioBank here described consists of approx 30% of participants of non-European ancestry out of 174,712 participants. Although other projects are ongoing to increase diversity in human genetic studies (like All of Us, Uk Biobank, Genome Asia, H3Africa, and TOPMed), there is still so much to be done in order to compensate for the absence of diversity. Anurag Verma et al work has a high representation of African-American patient participants in and addresses the urgent need to improve the quality of health care for all.
My advice is to publish the paper in its current form
Response 1: Thank you for reviewing our manuscript and providing thoughtful comments.
Round 2
Reviewer 1 Report
I'd like to thank the authors for a responsive revision, which has addressed all my concerns. I have a few remaining minor comments:
1) The paper now mentions ~44k participants with genomic data, but an additional 8,595 subjects were GWASed separately. It's not clear why these additional 8,595 subjects are not mentioned in eg Table 1 and first paragraph of discussion? That is only the 44k participants are mentioned.
2) Table S1 had "NULL" in cell C3, please correct.
Author Response
Response to Reviewer 1
Point 1: The paper now mentions ~44k participants with genomic data, but an additional 8,595 subjects were GWASed separately. It's not clear why these additional 8,595 subjects are not mentioned in eg Table 1 and first paragraph of discussion? That is only the 44k participants are mentioned.
Response 1: Thank you for your comment. Additional genotyping was performed on 8,595 individuals who were part of 44,000 participants in PMBB. We have revised the text on line 152 to clarify it.
“In addition, approximately 8,595 of the 44,000 PMBB participants were also genotyped in the Center for Applied Genomics (CAG) at Children’s Hospital of Philadelphia on the GSAv1 and GSAv2 genotyping array.”
Point 2: Table S1 had "NULL" in cell C3, please correct.
Response 2: We have fixed the issue.